# The Curious Role of PAI-1 in Severe Obstructive Sleep Apnea

**DOI:** 10.3390/biomedicines12061197

**Published:** 2024-05-28

**Authors:** Tea Friščić, Edvard Galić, Domagoj Vidović, Petrana Brečić, Igor Alfirević

**Affiliations:** 1Clinical Hospital Sveti Duh, 10000 Zagreb, Croatia; edvardgalic1@gmail.com (E.G.); alfirevic.igor@gmail.com (I.A.); 2School of Medicine, University of Zagreb, 10000 Zagreb, Croatia; petrana.brecic@bolnica-vrapce.hr; 3University Psychiatric Hospital Vrapče, 10000 Zagreb, Croatia; domagoj.vidovic@bolnica-vrapce.hr

**Keywords:** plasminogen activator inhibitor-1, obstructive sleep apnea, continuous positive airway pressure, atherogenesis, fibrinolytic capacity

## Abstract

Plasminogen activator inhibitor-1 (PAI-1) has a significant role in fibrinolysis, atherogenesis, cellular senescence, and chronic inflammation. OSA (obstructive sleep apnea) leads to increased PAI-1 levels and the development of cardiovascular disease (CVD). The aim of this study was to determine the effects of CPAP therapy on coagulation parameters and PAI-1 in patients with severe OSA. This prospective, controlled study enrolled 57 patients who were newly diagnosed with severe OSA, 37 of whom had had good CPAP adherence after 6 months of therapy (usage of the device for at least 4 h per night), and their data were analyzed. The analysis showed a statistically significant increase in D-dimer values before CPAP therapy (415 (316.5–537.5)) vs. after therapy (499 (327–652)), *p* = 0.0282, and a decrease in fibrinogen values (3.665 ± 0.752 before CPAP therapy vs. 3.365 ± 0.771 after therapy, *p* = 0.0075)). PAI-1 concentration values before and after CPAP therapy did not differ significantly (17.35 ± 7.01 ng/mL before CPAP therapy vs. 17.42 ± 6.99 ng/mL after therapy, *p* = 0.9367). This study shows a tendency for fibrinolytic capacity to improve in patients with OSA after CPAP therapy, although PAI-1 levels did not differ significantly.

## 1. Introduction

Plasminogen activator inhibitor-1 (PAI-1) is a serine protease inhibitor which has a significant role in fibrinolysis as it interferes with plasminogen activation by inhibiting tissue-type PA (tPA) and urokinase-type PA (uPA) [1].

PAI-1 is synthesized in an active form but is spontaneously converted to a latent form which has a half-life of two to three hours and is excreted via the liver. The concentration of circulating PAI-1 in its active form is 5–50 ng/mL, with large intra- and interpersonal differences and circadian fluctuations [2].

Research conducted over the past two decades showed that PAI-1 plays a role in regulating mechanisms and participating in processes that are not directly related to antifibrinolytic activity. These include the regulation of atherosclerotic processes, neointimal hyperplasia, cell migration, myoendothelial junction, skeletal muscle injury response, insulin signaling, obesity, Alzheimer’s disease, multiple sclerosis, cellular senescence, cancers, and the inflammatory response to ischemia and reperfusion [3]. Elevated PAI-1 levels have been found in patients with myocardial infarction, stroke, type 2 diabetes, insulin resistance, and metabolic syndrome [4,5].

Recent research has shown that PAI-1 is not only expressed in endothelial and smooth muscle cells (vasculature) but also in skeletal muscle cells, immune cells, the heart, the liver, the kidneys, adipose tissue, and some cancers. The transduction of Serpine1 (the gene encoding PAI-1) and the release of PAI-1 are induced by profibrotic (TGF-β) and proinflammatory (TNF-α; interleukins) factors and hormonal signals (insulin, IGF-1, glucagon, and cortisol) [6]. In addition, the promoter region of PAI-1 has an element that is sensitive to glucocorticoids and a site that responds to the presence of aldosterone by increasing the expression of PAI-1 [4]. Reactive oxygen species (ROS) influence the transcription of the Serpine1 gene by signaling via TGF and simultaneously stimulating activator protein-1 (AP-1), hypoxia-inducing factor-1 (HIF-1), and p53. TGF signaling also leads to a longer half-life of the mRNA for the Serpine1 gene and thus increases the translation of PAI-1 [6].

Obstructive sleep apnea (OSA) is a sleep-related breathing disorder resulting from repetitive upper airway collapse leading to intermittent hypoxemia, intrathoracic pressure fluctuations, and arousal states that result in autonomic nervous system (ANS) dysregulation, oxidative stress, low-grade chronic inflammation, and metabolic changes that promote endothelial dysfunction and ultimately lead to an increased risk of vascular events [7].

As shown in previous studies, OSA is associated with hypercoagulability and a disturbed coagulation–fibrinolysis balance due to changes in coagulation factors such as D-dimers, fibrinogen, and PAI-1 and enhanced platelet activation and aggregation. However, the relationship between OSA and some individual coagulation factors or regulator molecules, as well as the effect of CPAP therapy on hemostasis, is inconclusive [7].

OSA leads to increased PAI-1 levels and the development of cardiovascular disease (CVD) via multiple mechanisms, including increased ROS levels, the stimulation of the inflammatory milieu, concomitant insulin resistance and obesity, hypoxia, fibrosis, and arterial hypertension. The increased synthesis and secretion of PAI-1 lead to endothelial dysfunction and atherogenesis via mechanisms of further stimulation of inflammation, the inhibition of endothelial nitric oxide synthase (eNOS), neointimal hyperplasia, senescence, and hypofibrinolysis [2].

Although patients with OSA have chronically elevated PAI-1 levels, PAI-1 as a biomarker or promoter of CVD in these patients has not been adequately studied in clinical practice. Considering that elevated PAI-1 levels are an independent risk factor for CVD and that PAI-1 levels are elevated in patients with OSA, as well as the fact that the current treatment of OSA with CPAP therapy has not resulted in a reduction in cardiovascular risk according to large studies [8], it is challenging to investigate the impact of CPAP therapy on PAI-1 values, especially in the long term. Previous studies have not provided clear results on PAI-1 levels after CPAP therapy [9,10,11]. Could PAI-1 be the missing piece of the puzzle between OSA, CVD, and CPAP therapy?

## 2. Materials and Methods

### 2.1. Patient Selection and Study Design

This study was designed as a prospective, controlled, and cohort-based study.

All patients who were newly diagnosed with severe OSA (AHI ≥ 30) after an overnight hospital polysomnography (PSG) at a clinic for sleep disorders during the period from July 2019 to November 2020 were included in the study if they did not meet any of the exclusion criteria (pregnancy, severe chronic obstructive pulmonary disease (COPD), severe chronic renal insufficiency, atrial fibrillation, acute heart failure, cerebrovascular insult or transient ischemic attack in the last 6 months, acute coronary syndrome in the last 6 months, and chronic anticoagulant therapy). PSG was performed with a Respironics Alice^®^ 6: Sleepware G3 3.7.3 device, Philips, Murrysville, PA, USA. The results were interpreted according to the American Academy of Sleep Medicine’s rules for scoring respiratory events [12]. All data were stored on a computer, manually scored, and evaluated by a certified sleep physician and technician. Apnea was considered the complete cessation of breathing, i.e., a signal reduction by ≥90%, for ≥10 s. Hypopnea was considered partial respiratory arrest, i.e., a signal reduction ≥ 30%, lasting ≥10 s with a drop in oxygen saturation by ≥3% or arousal.

This study was conducted in accordance with the Declaration of Helsinki and approved by the local ethics committee. All patients signed a previously explained informed consent form in order to participate in this study. The first study visit was performed before the patients began receiving CPAP therapy. At the first visit, the patient’s medical history and status were recorded, as were their anthropometric measurements (height, weight, waist, and neck circumference), and an Epworth sleepiness scale (ESS) questionnaire was administered. Of the 58 patients who initially entered, 57 were enrolled, i.e., did not have any of the exclusion criteria after the initial cardiovascular assessment (one had newly diagnosed atrial fibrillation). Blood sampling for laboratory analysis was carried out in the morning (between 7 and 11 am) on an empty stomach after at least 12 h of fasting. Samples were analyzed within 30 min of venipuncture except for the samples used for a PAI-1 analysis, which were centrifuged (1000 revolutions/min for 15 min) and frozen at a temperature of ≤−20 °C. D-dimer was analyzed via immunoturbidimetric analysis (BCS XP, Siemens Healthcare GmbH, Erlangen, Germany), fibrinogen levels were analyzed using the modified Clauss method (BCS XP, Siemens Healthcare, Erlangen, Germany), and PV, INR, and APTT were analyzed by coagulometry (BCS XP, Siemens Healthcare GmbH, Erlangen, Germany).

CPAP devices from two manufacturers were used (prisma Smart, Loewenstein Medical Technology GmbH Co. KG, Hamburg, Germany; S9 Escape, ResMed, San Diego, CA, USA). CPAP treatment was administered during hospitalization, and the titration pressure assumed to be effective at treating the majority of events was defined at the 95th percentile. The follow-up study visit occurred after a minimum of 6 months of CPAP device usage and included the same assessment as the first visit. Of the 57 enrolled patients, 37 had good CPAP adherence (usage of the device for at least 4 h per night; data were obtained from each device’s memory card), and their data were evaluated in this study. After the collection of all samples, an enzyme-linked immunosorbent assay (ELISA) was performed to determine the value of PAI-1, using a Human Serpin E1/PAI-1 kit (Quantikine ELISA, Catalog Number DSE100) according to the manufacturer’s instructions. This study is an extension of a previously published study which used the same group of patients [13].

### 2.2. Statistical Analysis

The required number of subjects, which was determined using a paired-samples *t*-test with a test power of 80% and a significance level of 0.05, was 34, with an expected 25% decrease in PAI-1 after 6 months of therapy with the CPAP device compared to baseline values (the analysis was performed using G*Power for Windows, version 3.1.9.2).

A normality analysis of the distribution of the numerical data was performed using the Shapiro–Wilk test, and appropriate parametric and/or non-parametric statistical methods were applied depending on the results obtained. Quantitative data are represented by ranges, arithmetic means, and standard deviations in the case of a parametric distribution or by medians and interquartile ranges in the case of a non-parametric distribution. Categorical data are represented by numbers (percentages). Differences in quantitative values between the individual measurements were analyzed using the dependent *t*-test or the Wilcoxon test. All *p*-values less than 0.05 were taken into account. The Python version 3.8 programming language was used for the analysis.

## 3. Results

The data of 37 patients who had satisfactory adherence, i.e., average use of the device for more than 4 h during the night after at least 6 months of CPAP therapy, were included in the further analysis. The demographic and clinical characteristics of the subjects are listed in Table 1.

The average age of the patients was 53 ± 10 years with a BMI in the obese range (average 34.4 ± 6.1 kg/m^2^). The majority were male (78%), and 27% of them were smokers. The most common comorbidity was arterial hypertension (51%), followed by diabetes (16%). The patients generally had no previous CVD.

The polysomnographic parameters, assessment of sleepiness, and duration of CPAP therapy are listed in Table 2.

The average AHI was 58.4 ± 22. Polysomnography showed a predominantly obstructive component in the apnea episodes (OA 33.35 vs. CA 1.3). The mean duration of apnea was 24.2 ± 6.1 s, and the mean lowest measured oxygen saturation during sleep was 74.03 ± 11.36%. The mean Epworth Sleepiness Scale score was 10.6 ± 5.2, and the mean duration of CPAP therapy was 290.49 ± 56.7 days. The average time of CPAP therapy during the night was (322.3 ± 51.3 min).

A coagulogram and the PAI-1 parameters before and after CPAP therapy are shown in Table 3.

An analysis of the values of the coagulation components showed a statistically significant increase in INR (0.94 (0.91–0.97) before CPAP therapy vs. 0.98 (0.93–1) after therapy, *p* = 0.0033) and D-dimer values (415 (316.5–537.5) before CPAP therapy vs. 499 (327–652) after therapy, *p* = 0.0282) and a decrease in fibrinogen values (3.665 ± 0.752 before CPAP therapy vs. 3.365 ± 0.771 after therapy, *p* = 0.0075). The PAI-1 concentration values before and after CPAP therapy did not differ significantly (17.35 ± 7.01 ng/mL before CPAP therapy vs. 17.42 ± 6.99 ng/mL after therapy, *p* = 0.9367). Normalization of the PAI-1 concentration with total proteins and albumins (PAI-1 concentration/total proteins, i.e., PAI-1 concentration/albumin) was also performed in order to eliminate the possible impact of hypovolemia or hypervolemia on the results, but there was still no statistically significant difference (normalization with total proteins 0.24 ± 0.1 before and after therapy, *p* = 0.6638; normalization with albumin 0.41 ± 0.17 vs. 0.4 ± 0.16, *p* = 0.6376).

Anthropometric measurements, blood glucose measurements, and AHI and ESS scores before and after CPAP therapy are shown in Table 4.

During the study period, anthropometric measurements including weight, waist and neck circumference, and BMI did not change significantly. Blood glucose measurements including fasting plasma glucose (FPG) and glycosylated hemoglobin (HbA1c) also did not significantly change. AHI and ESS values showed a statistically significant decrease (58.4 ± 22 before CPAP therapy vs. 4 (2.9–5) after CPAP therapy, *p* = 0, and 10.6 ± 5.2 before CPAP therapy vs. 4 (3–6) after CPAP therapy, *p* = 0, respectively) which was expected in patients with good therapy adherence.

## 4. Discussion

The link between OSA and CVD is well known and has been confirmed by numerous studies over several decades. The pathophysiological mechanisms leading to the development and increased incidence of CVD in patients with OSA involve several interrelated cascade processes. The primary event that triggers all cascade processes is a complete or partial collapse of the upper airway leading to gas exchange disturbances, changes in intrathoracic pressure, and sleep fragmentation. The basis of treatment for severe OSA is the use of CPAP, which suppresses this primary event and allows for unobstructed airway passage. Since the treatment eliminates the primary cause, it would be logical to assume that the treatment also reduces the risk of CVD, i.e., events. Observational studies have confirmed a statistically significant reduction in endothelial dysfunction [14] as well as lower incidences of death, myocardial infarction, and stroke in patients with OSA treated with CPAP [15]. However, randomized clinical trials, which are the gold standard of evidence-based medicine, showed no significant improvement in cardiovascular outcomes following the use of CPAP therapy [8,16,17]. There are several possible reasons that have led to statistically insignificant results in large randomized trials. One limitation common to all studies is treatment adherence, which may not have been sufficient to detect differences in outcomes. A meta-analysis that included four randomized trials with more than 3700 subjects showed a statistically significant reduction in the risk of MACE in the subgroup of subjects who used a CPAP device for an average of more than 4 h during the night, suggesting that adherence to CPAP therapy is crucial for improving cardiovascular outcomes [18].

The aim of this study was to determine the effects of CPAP therapy on coagulation parameters and PAI-1. In previously published studies comparing the concentration of PAI-1 before and after CPAP therapy, the follow-up period was relatively short, i.e., a few weeks, and this study is the first to follow the change in PAI-1 concentration over a longer period of at least 6 months [9,10,11,19]. In addition, only the results of patients who demonstrated good adherence to therapy were included in this study as adherence to therapy was one of the main limitations in previous studies.

Several intermediary mechanisms involved in the development of CVD in OSA are closely intertwined with the regulation of the fibrinolysis system, in which PAI-1 is an important carrier. This study hypothesized a decrease of at least 25% in PAI-1 levels after the use of CPAP. A statistical analysis of the data obtained showed that the difference in PAI-1 concentration before and after CPAP therapy was not statistically significant, even after the values were normalized by the concentrations of total proteins and albumin. The coagulation parameters that achieved a statistically significant difference were the D-dimer (increase) and fibrinogen (decrease) values.

Phillips et al. were also unable to demonstrate a statistically significant difference in PAI-1 levels, fibrinogen, and D-dimer in 28 subjects with severe OSA after two months of CPAP therapy compared to the placebo group [9]. However, another group of authors showed in a post-hoc analysis a significant decrease in PAI-1 concentration after 2 weeks of CPAP therapy in 44 subjects [11] but without significant differences in the values of other parameters, e.g., D-dimer. The results of a study by Steffanina et al. showed a statistically significant reduction in PAI-1 levels after one month of CPAP therapy in patients with OSA compared to a control group [10].

One of the possible explanations of the results of this study may be that the short-term beneficial effects of CPAP on PAI-1 values are reversed by homeostatic factors in the long term [7].

The results of this study showed a statistically significant increase in D-dimer levels after CPAP therapy which were still in the normal range, i.e., less than 500 ng/mL. The reduced fibrinolytic capacity in patients with OSA is partly a combination of increased levels of PAI-1 and decreased levels of D-dimer. The role of D-dimer is usually understood as a marker of fibrin formation leading to atherothrombotic events, but D-dimer levels also reflect fibrin degradation, suggesting that reduced D-dimer levels could reflect reduced fibrinolytic potential [7,20]. From this standpoint, an increase in D-dimer levels without exceeding the normal range could be a signal of enhanced fibrinolytic capacity in OSA patients when using CPAP therapy.

The level of fibrinogen decreased statistically significantly after CPAP therapy, suggesting a possible reduction in coagulability but also in the intensity of chronic inflammation, considering the role of fibrinogen in inflammatory events. Elevated fibrinogen levels increase the risk of thrombosis due to increased blood viscosity, which leads to slower blood flow and the aggregation of erythrocytes and platelets [21]. Studies have shown that elevated fibrinogen levels are associated with the risk of developing CVD [22].

Vascular senescence, which is characterized by growth arrest and alterations in the gene expression profiles of endothelial cells and vascular smooth muscle cells, is another pathological mechanism by which PAI-1 affects the development of CVD [23], thus raising the question as to whether PAI-1 and vascular senescence could be the answer to why CPAP therapy does not improve cardiovascular outcomes. Also, senolytic PAI-1 targeted therapy may be beneficial in CVD development.

Taking into account the significant increase in D-dimer levels and the decrease in fibrinogen levels in patients with OSA after CPAP therapy, this study shows a tendency to improve fibrinolytic capacity, although PAI-1 levels did not differ significantly. The reason for this could be the multiple associations of PAI-1 with different inflammatory and fibrinolytic mechanisms. The influence of chronic inflammation on PAI-1 levels in patients with obesity and diabetes mellitus had no significant impact on the results of this study as BMI and HbA1c levels did not change significantly during the study period. The change in PAI-1 perhaps requires a longer duration of therapy. Also, due to the strong intertwining of PAI-1 with other mechanisms of the coagulation cascade and chronic inflammatory response, the acute change could be reversed by homeostatic factors in the long term. It is possible that PAI-1 plays a greater role in advanced atherosclerosis and that its levels did not decrease due to the characteristics of the patients, who were relatively young compared to the general population and had a low number of comorbidities. PAI-1 levels are different in serum and plasma, considering that a significant portion of PAI-1 is bound to platelets. Other pre-analytical factors that could have influenced the PAI-1 concentration and led to this result but were not taken into account are smoking, physical activity, and the fat content of the diet.

The limitation of this study is primarily its small number of subjects which, although it met statistical power for changes in PAI-1 concentration, may have influenced the results. The cohort sample is not fully generalizable to the general population. The subjects were predominantly male and had a relatively low number of comorbidities. In addition, the subjects received individualized chronic therapy, some of which was changed during follow-up. Future research in this area with a larger and more representative cohort and a longer follow-up period could provide new insights.

## 5. Conclusions

OSA leads to increased PAI-1 levels and the development of cardiovascular disease through multiple mechanisms, including changes in the coagulation cascade and supporting the chronic inflammatory response. This study shows a tendency to improve the fibrinolytic capacity in patients with OSA after CPAP therapy.

## Figures and Tables

**Table 1 biomedicines-12-01197-t001:** Demographic and clinical characteristics of the study group.

Age (years)	53 ± 10
Male/female gender	29/8
BMI (kg/m^2^)	34.4 ± 6.1
Smoker (number/%)	10/27
Comorbidities	
Hypertension (number/%)	19/51
CVD	1 *
COPD	2
Diabetes mellitus (number/%)	6/16
CKD	1 *

BMI—body mass index; CKD—chronic kidney disease; CVD—cardiovascular disease including acute coronary syndrome; stroke/transient ischemic attack/heart failure; COPD—chronic obstructive pulmonary disease. * Only one patient had a history of heart failure and mild CKD.

**Table 2 biomedicines-12-01197-t002:** Polysomnographic parameters, assessment of sleepiness, and duration of CPAP therapy of the study group.

Polysomnographic parameters	
AHI (events/h)	58.4 ± 22
CA (events/h)	1.3
OA (events/h)	33.35
Hypopnea (events/h)	11.15
Average apnea duration (s)	24.2 ± 6.1
Minimum O_2_ saturation (%)	74.03 ± 11.36
ESS	10.6 ± 5.2
Average time of CPAP therapy per night (min)	322.3 ± 51.3
Duration of CPAP therapy (days)	290.49 ± 56.7

AHI—apnea–hypopnea index; CA—central apnea; CPAP—continuous positive airway pressure; ESS—Epworth sleepiness scale; OA—obstructive apnea.

**Table 3 biomedicines-12-01197-t003:** Coagulogram and PAI-1 parameters before and after CPAP therapy.

	Before CPAP Therapy	After CPAP Therapy	*p* Value
Coagulogram			
PT	1.15 (1.08–1.22)	1.12 (1.06–1.19)	0.3337
INR	0.94 (0.91–0.97)	0.98 (0.93–1)	0.0033
APTT	23.15 (21.95–24.6)	23.45 (22.5–24.85)	0.4511
D-dimer	415 (316.5–537.5)	499 (327–652)	0.0282
Fibrinogen (g/L)	3.665 ± 0.752	3.365 ± 0.771	0.0075
PAI-1			
PAI-1 (ng/mL)	17.35 ± 7.01	17.42 ± 6.99	0.9367
PAI-1 norm. protein	0.24 ± 0.1	0.24 ± 0.1	0.6638
PAI-1 norm. albumin	0.41 ± 0.17	0.4 ± 0.16	0.6376

APTT—activated partial thromboplastin time; CPAP—continuous positive airway pressure; INR—international normalized ratio; PAI-1—plasminogen activator inhibitor-1; PT—prothrombin time.

**Table 4 biomedicines-12-01197-t004:** Anthropometric measurements, blood glucose measurements, AHI and ESS before and after CPAP therapy.

	Before CPAP Therapy	After CPAP Therapy	*p* Value
Anthropometric measurements			
Weight (kg)	105.92 ± 23.37	103.97 ± 20.25	0.1268
Waist circumference (cm)	116.16 ± 16.44	115.19 ± 14.19	0.4057
Neck circumference (cm)	45.65 ± 4.6	45.46 ± 4.5	0.4814
BMI (kg/m^2^)	34.4 ± 6.1	33.835 ± 5.163	0.1107
Blood glucose measurements			
FPG (mmol/L)	6.1 (5.5–6.8)	6.3 (5.7–6.8)	0.6804
HbA1c (%)	6 (5.7–6.5)	6.3 (5.8–6.6)	0.1435
AHI	58.4 ± 22	4 (2.9–5)	0
ESS	10.6 ± 5.2	4 (3–6)	0

AHI—apnea–hypopnea index; BMI—body mass index; CPAP—continuous positive airway pressure; ESS—Epworth sleepiness scale; FPG—fasting plasma glucose; HbA1c—glycosylated hemoglobin.

## Data Availability

The data presented in this study are available upon request from the corresponding author. The data are not publicly available due to privacy restrictions.

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
