# Peer review of "The Curious Role of PAI-1 in Severe Obstructive Sleep Apnea"

_biomedicines, 2024, doi:10.3390/biomedicines12061197_

Round 1

Reviewer 1 Report

Comments and Suggestions for Authors

The research conducted by Friscic et al. highlights the positive impact of consistent CPAP adherence on fibrinolytic capacity among patients with OSA. While the study revealed no significant change in PAI-1 levels before and after treatment, it contributes to the existing body of literature by extending the duration of CPAP assessment. Although similar assessments have been conducted previously, this study stands out as one of the pioneering efforts to evaluate PAI-1 and fibrinolysis activity over a more prolonged period of CPAP treatment.

Minor comment:

1) Should have the samples been frozen at -80°C instead of -20°C, could  that have affect results of plasma PAI-1?

2) Is it feasible to categorize adherent patients into two groups: the top 50% with maximal CPAP adherence versus the lower 50% with normal acceptable adherence? Furthermore, does maximal CPAP adherence versus standard adherence levels affect PAI-1 levels or fibrinolytic activity?

3) Discussion section is a bit confusing. It would be beneficial for the authors to delve deeper into their findings, particularly the contentious increase in d-dimer levels. For instance, normal d-dimer levels are less than 500ng/ml. even if its statistically significant after CPAP adherence, it is still not considered as a positive value. Additionally, expanding the Discussion to encompass 3-4 paragraphs would enhance clarity and thoroughness.

4) A limitation section should be added. 

Reviewer 2 Report

Comments and Suggestions for Authors

I think this article is very well written, the research problem statement is interesting, the research design is rigorous, and the discussion and explanation of the results are quite clear, however, I have some comments as follows:

1. PAI-1, D-dimer, and fiblinogen are factors of the fibrinolytic system, so their changes may influence each other. Please add a discussion on this.

2. There is no discussion of descriptive research results such as BMI, ESS, and blood glucose before and after 6 months of CPAP treatment. These parameters can affect PAI-1, D-dimer, and fiblinogen.

3. The author mentions that adherence to CPAP treatment is important in influencing outcomes, but there is no discussion about whether it improves BMI and ESS, and what its impact is on outcomes.

5. Spelling error: repeating the word "significantly" in line 237.

Round 2

Reviewer 2 Report

Comments and Suggestions for Authors

This manuscript has been suitably revised. I recommend that it would be accepted.